# Effects of Hydrothermal Processing on Volatile and Fatty Acids Profile of Cowpeas (*Vigna unguiculata*), Chickpeas (*Cicer arietinum*) and Kidney Beans (*Phaseolus vulgaris*)

**DOI:** 10.3390/molecules27238204

**Published:** 2022-11-24

**Authors:** Prit Khrisanapant, Biniam Kebede, Sze Ying Leong, Indrawati Oey

**Affiliations:** 1Department of Food Science, University of Otago, Dunedin 9054, New Zealand; 2Riddet Institute, Palmerston North 4442, New Zealand

**Keywords:** legumes, hydrothermal processing, volatile compounds, fatty acid, foodomics, chemometrics, lipid oxidation, Maillard reaction

## Abstract

Legumes are an economical source of protein, starch, dietary fibre, fatty acids, vitamins and minerals. However, they are not as fully utilised, due to volatile compounds contributing to their undesirable odour. The purpose of this work was to understand the processing time’s effect on the legumes’ volatile profile. Hence, this study investigated the effects of hydrothermal processing times on the volatile and fatty acids profiles of cowpeas, chickpeas and kidney beans. All legumes were pre-soaked (16 h) and then hydrothermally processed at 95 °C for 15 to 120 min, using an open system to approximate standard household cooking practices and a closed system to represent industrial processing. Alcohol, aldehyde, acid and ester volatile compounds showed decreasing trends during processing, which can be associated with enzyme inactivation and process-induced degradation. This work showed that processing at 95 °C for 30 min significantly reduced the number of compounds commonly associated with undesirable odour, but showed no significant change in the fatty acid profile. Other volatiles, such as furanic compounds, pyrans and sulphur compounds, showed an increasing trend during processing, which can be related to the Maillard reactions. This observation contributes to the growing knowledge of legume processing and its impact on volatile flavour. It can advise consumers and the industry on selecting processing intensity to maximise legume utilisation.

## 1. Introduction

Legumes have been utilised by civilisation since the beginning of agriculture. They are an economical source of carbohydrates, minerals and plant-based protein [1]. Legumes’ long shelf life (via dry seed storage) also means they can contribute to food security. Unfortunately, legumes are not as fully utilised, partly due to the volatile compounds contributing to their odour being considered undesirable by the vast proportion of people [2], flatulence and lower protein digestibility (compared to animal proteins). The lower protein digestibility is due to natural barriers in the plant cell and antinutrients such as enzyme inhibitors, tannins and phytic acid [3,4]. Furthermore, from a consumer’s perspective, legumes require time-consuming preparation, mainly if it exhibits a hard-to-cook effect. Thus, legumes are commonly processed (i.e., soaking, boiling) before consumption, both to increase their palatability and inactivate deleterious antinutrients to improve digestibility [4,5,6,7]. A key avenue for increasing legumes’ palatability is via adequate food processing to minimize the formation of undesirable odour-active volatile compounds, such as through hydrothermal processing, drying techniques or even plant variety breeding efforts [8,9]. Nevertheless, there is still a limited understanding of the impact of hydrothermal processing on legumes’ volatile profiles.

Volatile compounds in legumes are key contributors to aroma [2]. Depending on their concentration and interaction with other compounds, some volatile compounds, such as hexanal and 1-octen-3-ol, can contribute to the undesirable odours of (un)cooked legumes [2]. In legumes, the enzymatic and non-enzymatic oxidation of lipids is a major contributor to their volatile profiles [10,11]. Furthermore, thermal processing can trigger the Maillard reaction, Strecker degradation and other thermal degradation reactions leading to the formation of several odour-active volatile compounds (e.g., furans, Strecker aldehydes, sulphur compounds, etc.). Therefore, changes in the volatile compounds can be used as a testimony to the effects of processing on these complex reactions and their overall flavour. However, there is a lack of studies systematically investigating the impact of different hydrothermal processing intensities on the volatile compounds in commonly consumed legumes, such as cowpeas, chickpeas and kidney beans, since the current literature has focused mainly on soybeans [10,12,13,14,15,16,17]. For cowpeas, a study was conducted to investigate their volatile profile in the dried state [18]. For chickpeas, the effect of roasting on chickpeas’ volatile profile has been considered [19,20], though only one study has investigated hydrothermal processing (but only at one processing intensity) [21]. Similarly, most reported studies focus on the dried state for chickpeas’ fatty acid profile [22]. Likewise, information regarding the effects of hydrothermal processing on kidney beans’ fatty acid profile is scarce. Lastly, even though the volatile profile of kidney beans has been studied in both hydrothermally processed and dried states in previous studies [23,24,25], a simulated home-cooking process, as would be reasonably experienced by a typical consumer, has not been explored.

According to our knowledge, there are no studies systematically investigating different boiling durations (processing intensities) on the volatile profile of legumes, despite boiling being a common processing method. Secondly, no studies are researching and comparing the volatile compounds under the home (opened) cooking versus sealed (closed) conditions. Furthermore, the lack of publications investigating fatty acids and volatile profile changes concurrently is an opportunity, as lipid oxidation plays a crucial role in forming odour-active volatile compounds, such as aldehydes, alcohols and ketones [10]. Thirdly, most studies in the literature focused only on a priori selected few volatile compounds; thus, there is a need for a more comprehensive untargeted fingerprinting approach to cover a broad range of volatile flavour compounds.

The objective of this paper is to study the effects of hydrothermal processing on cowpeas, chickpeas and kidney beans, where each legume is studied step-by-step in both an approximation of home cooking (referred to as opened system) and an approximation of industrial cooking (referred to as closed system) of different processing time (up to 120 min). The impact of the process will be investigated on the fatty acid and volatile profiles, and an untargeted GC-MS-based fingerprinting method was implemented to monitor the evolution of an increased number of volatile compounds during processing. Chemometrics was employed to understand the key patterns and identify discriminant compounds.

## 2. Results and Discussions

### 2.1. Starch, Protein, Lipid and Moisture Content of Cowpea, Chickpea and Kidney Bean

The cowpeas used in this study constituted 49.49%, 21.58%, 3.46% and 9.13% of starch, protein, lipid and moisture, respectively. The chickpeas constituted 47.25%, 16.74%, 7.73% and 8.27% of starch, protein, lipid and moisture, respectively. The kidney beans constituted 51.07%, 16.39%, 3.59% and 12.52% of starch, protein, lipid and moisture, respectively. The estimated values are comparable to levels reported in the literature, where cowpeas have a high protein and low-fat composition [26], chickpeas have a medium protein and high fat composition [27] and kidney beans have a medium protein and low-fat composition [28].

### 2.2. Lipid Yield and Relative Fatty Acid Composition of Cooked Cowpea, Chickpea and Kidney Bean

The lipid yields and relative proportions of fatty acids in legumes are presented in Table 1. Hydrothermal processing (of any duration) had a negligible effect on the lipid yield obtained in chickpeas [29] and kidney beans, suggesting only a slight change in the content or extractability of lipids.

The lipid yield of cowpea progressively decreased as a function of *t_c_*, with the greatest changes occurring in the first 30 min. Compared to the yield of the soaked sample (*t_c_* = 0 min), processing cowpeas for 15 min or more reduced the lipid yield by 15.1% to 27.8% (significant at *p* = 0.05). A previous study with hydrothermal processing of mung beans also reported a decrease in phospholipids and triglycerides, hypothesised to be converted into monoglycerides, 1,2-(2,3)-diglycerides, sterols and free fatty acids [30]. It is possible that lipid in cowpeas was susceptible to thermal- and water-catalysed oxidation [31].

Analysis of fatty acid methyl esters detected five clearly separated peaks in the cooked legume samples representing palmitic (C16:0), stearic (C18:0), oleic (C18:1), linoleic (C18:2) and α-linolenic (C18:3) acids. The proportions of these fatty acids are shown in Table 1. Overall, the effects of processing on the fatty acid profile are limited, with no clear trend. A previous study on hydrothermal processing (45 min boiling) and fermentation showed that the proportion of fatty acid in cowpeas remained unchanged [32]. Therefore, it may be that any oxidation of fatty acid is not major enough during the experimental timeframe to show a difference that is detectable by GC-FID. Though are no previous studies on the effects of hydrothermal processing on the lipid profile of chickpeas and kidney beans, the proportional values obtained in this investigation are comparable to chickpeas’ and kidney beans’ unprocessed state as obtained by Caprioli, Giusti, Ballini, Sagratini, Vila-Donat, Vittori and Fiorini [33]. However, the fatty acid profile would still play a role in the volatile formation, since a small amount is sufficient to promote oxidative reactions that trigger changes in the volatile compounds [10]. Thus, the role of lipid oxidation cannot be discounted [11].

### 2.3. Untargeted Headspace Volatile Fingerprinting of Cooked Legumes

Volatile analysis of legumes processed via the opened system resulted in the detection and tentative identification of 74, 62 and 53 volatile compounds across all treatments in cowpeas, chickpeas and kidney beans, respectively (Figure 1). For the closed system, the total number of detected volatiles in cowpeas is comparable at 70 and to detected volatile compounds in chickpeas, while kidney beans have increased to 77 and 69 compared to the open system (Figure 1). The detected volatile compounds consisted of acids, alcohols, aldehydes, alkanes, alkenes, esters and lactones, furans, hydrocarbons, ketones, pyrans, pyrazines, terpenes and sulphur-containing compounds. The detected volatile classes are comparable to those reported in the literature [18,19,20,21,23,25].

Furans and sulphur-containing volatile compounds are two chemical classes detected in greater numbers in the closed system for all legumes (Figure 1). This could be because conducting hydrothermal processing of cowpeas, chickpeas and kidney beans in a closed system retained volatile compounds that would have otherwise escaped into the environment during processing.

The detected volatile chemical classes in cowpeas consisted of alcohol, terpene and aldehyde, such as 1-hexanol, hexanal and linalool. With 74 and 70 detected volatiles, respectively, in the opened and closed systems, this is an improvement compared to 27 volatile compounds detected by Fisher, Legendre, Lovgren, Schuller and Wells [18] in dried, grounded cowpeas through carrier gas sweeping and detection with GC-MS.

In chickpeas, the most abundant volatiles comprised aldehydes, alcohols and ketones, such as nonanal, 1-penten-3-ol and 3,5-octanedione. The study of Rembold, Wallner, Nitz, Kollmannsberger and Drawert [20] reported 154 volatile compounds in dried chickpeas, consisting of aliphatic hydrocarbon, terpene and alcohol. It is worth noting that a different volatile recovery method, the Tenax trap method, was employed by Rembold, Wallner, Nitz, Kollmannsberger and Drawert [20].

In kidney beans, alcohols, aldehydes, and ketones were the most represented chemical classes. With 53 and 69 detected volatile compounds in the opened and closed system, respectively, this was comparable to the study of Oomah, Liang and Balasubramanian [24], who reported 62 volatiles in dried Phaseolus beans. The study of Mishra, Tripathi, Gupta and Variyar [23] reported 79 volatile compounds in dried and cooked kidney beans; however, they used three different types of kidney beans, possibly explaining the difference.

The total ion chromatograms (TICs) of soaked and hydrothermally processed samples for each legume in the opened and closed system showed three key trends (Appendix A). Firstly, due to processing, there is an overall decrease in the abundance/intensity (*Y*-axis, Appendix A) of the most volatile compounds. Secondly, peaks not visible in the soaked samples were detected in abundance in cooked samples. Notably, in the closed system TICs, the peaks representing furan and benzopyran appeared after processing (Appendix A). Thirdly, the TICs obtained through the closed system approach showed increased abundance as a function of processing compared to the open system (Appendix A).

### 2.4. Investigating the Volatile Changes during Hydrothermal Processing Using Chemometrics

Headspace GC-MS fingerprinting was combined with chemometrics to discern significant trends/patterns in the legumes’ volatile compounds as a function of processing time. PCA was used as an exploratory technique to determine authentic trends, groupings and outliers. An apparent effect of processing on the volatile fractions was observed, and no outliers were detected (data not shown). Next, PLS-R models were constructed using three LVs (latent variables) for cowpeas and chickpeas and two LVs for kidney beans. The optimum number of LV for the PLS-R model was selected based on cross-validation, using the root mean squared error of the cross-validation. The bi-plots for cowpeas, chickpeas and kidney beans, constructed using the first two LVs, are presented in Figure 2. In each bi-plot, the coloured markers represent different sample treatments within a single legume type, and the small unfilled circles represent the volatile compounds. Samples positioned close to each other are considered similar (according to their volatile profile), and samples projected further from each other are considered to have a different volatile profile.

From all bi-plots, there is a clear separation in the soaked (*t_c_* = 0 min) compared to the processed (*t_c_* ≥ 15 min) samples represented in a V-shaped trend pivoting to the processed sample group (*t_c_* ≥ 15 min; Figure 2). This non-linear pattern suggests that multiple complex reactions could concurrently occur as a function of the processing time. As processing begins, the classes on the bi-plot pivot from the far left-middle of the bi-plot to just underneath the centre of the coordinate (*t_c_* = 15 min), indicating an apparent effect of processing, even with just 15 min of boiling. After that, as processing time increased towards 120 min, the samples projected further away from the origin into the upper right quadrant, increasing (positive) loading for both LV1 and LV2. Interestingly, this trend was consistent for all three legumes across both open and closed systems.

The distribution of the volatile compounds on the plot can be examined closer to monitor their change as a function of processing times. Volatile compounds positioned further away from the centre of the coordinate have a higher loading for the effect of time; therefore, processing plays a vital role in the observed classification. Indeed, many of them are projected away from the origin and positioned near the soaked samples, as well as a few towards the highest processing time (*t_c_* = 120). The close proximity of these compounds to a sample shows that they are found in greater quantities in those respective samples.

The two rings on the bi-plots show the confidence level of each volatile, contributing to the selection of LVs and classification/volatile changes. Specifically, those volatiles positioned between the inner and outer rings are within 70% to 100% confidence and are the compounds that change the most during processing. The closed system bi-plots (Figure 2) show more compounds within the two rings closer to the soaked sample, indicating the loss of an increased number of volatile compounds due to processing. This is the primary trend when visually investigating the bi-plots. Conversely, few volatile compound are projected in the direction of the processed samples, illustrating that a small number have increased or are formed due to hydrothermal processing. Overall, the bi-plot presents visual evidence of a clear pattern/trend due to hydrothermal processing. Next, the VID coefficients were calculated to rank the volatile compounds based on their importance for the classification. A high positive VID coefficient indicates an increase in the abundance of volatile compounds as a function of *t_c_*, whereas a negative coefficient indicates a decrease.

This study selected volatile compounds with an absolute VID coefficient higher than 0.600 as discriminant compounds, deriving the changes. This cut-off was confirmed with significance testing (analysis of variance; *p* = 0.05). The selected compounds are shown in Table 2 and Table 3. The selected discriminant compounds can be grouped into aldehyde, alcohol, acid, ester, ketone, terpene, hydrocarbon, furan and sulphur chemical classes.

### 2.5. Interpretation of Volatile Compounds Changing with Processing Time and Their Associated Reaction Pathways

Key patterns were discerned from the discriminant compounds in Table 2 and Table 3. Most of the compounds show a decreasing trend (selected with a negative VID coefficient), whereas few others show an increasing trend as a function of *t_c_*. The changes in these volatile compounds can be linked to two key reaction pathways: (i) enzyme-related oxidative reactions; and (ii) high-temperature-related degradative reactions.

#### 2.5.1. Enzyme-Related Reaction Pathways

In legumes, oxidative enzymatic reactions play a crucial role in changing a wide range of odour-active volatiles, such as aldehydes, alcohols, short-chain fatty acids and ketones. Aldehydes represent the majority of the selected discriminant compounds. These odour-active volatile compounds have significantly decreased during the first 15 min of boiling legumes in both open and closed systems (Figure 3A). The compounds 2(*E*)-nonenal, 2(*Z*)-heptenal and 2(*E*)-hexenal are specific examples of aldehydes, decreasing in cowpeas, chickpeas and kidney beans, respectively (Table 2 and Table 3). Hexanal, 2(*E*),4(*E*)-heptadienal are interesting, since they decrease in all three legumes. Mishra, et al. [23] observed a similar aldehyde reduction in red kidney beans using a simultaneous distillation extraction to simulate cooking. Likewise, Noordraven, Buvé, Chen, Hendrickx and Van Loey [21] also observed a decreased abundance of aldehydes in chickpeas cooked for 40 min compared to soaked chickpeas. In legumes, enzymes such as lipoxygenase are responsible for aldehyde formation through fatty acid oxidation (e.g., oleic, linoleic and α-linolenic acid). The species-specific isozymes of lipoxygenase, hydroperoxide lyase and isomerase can lead to the formation of a large variety of volatile aldehyde (Appendix A). Hydrothermal processing possibly inactivated these enzymes; hence the decrease observed in various aldehydes as a function of *t_c_*.

In the literature, the above volatile compounds are commonly associated with the grassy and fatty aromas in legumes [2]. Therefore, decreasing these odour-active volatiles could reduce the legumes’ beany and green odour [2]. Only one aldehyde, benzaldehyde, was increased during the closed system hydrothermal processing of all three legumes. However, the formation of this compound is commonly linked to different pathways; the Strecker degradation is a side reaction of the Maillard reaction and is mainly associated with high-temperature processing [34]. This trend was observable when a closed system approach was taken, wherein the highly volatile benzaldehyde was not lost into the environment.

Alcohols, such as 1-heptanol, 1-octen-3-ol, 1-hexanol and ethanol, are volatiles with the highest negative VID (most significant decrease as a function of *t_c_*) in cowpeas, chickpeas and kidney beans. Figure 3B shows a sharp reduction in 1-octen-3-ol during the first 15 min of the processing; a similar trend was observed for other alcohols, listed in Table 2 and Table 3. The literature reported a similar observation in cooked kidney beans [23] and chickpeas [21]. Comparable to the aldehydes, this could positively influence the overall flavour of the legumes, as alcohols such as 1-octen-3-ol possess an unpleasant musty odour with a low (1 ppb in water) odour threshold [35]. These volatile alcohol compounds generally evolved as enzymatically catalysed secondary degradation products of lipid oxidation in legumes (Appendix A). It can be hypothesized that those responsible enzymes are inactivated during boiling.

Hexanoic acid has significantly decreased during the hydrothermal processing of all three legumes. Hexanoic acid is a major product of the alcohol dehydrogenase pathway (Appendix A); where 1-hexanol may be converted into hexanoic acid, with Gomes, et al. [36] noting three alcohol dehydrogenase isozymes present in chickpeas. Therefore, it is likely that the hydrothermal processing halted the hexanoic acid production through the inactivation of alcohol dehydrogenase and other enzymes involved earlier in the pathway.

Some esters and lactones have decreased during the thermal processing of legumes. In cowpeas, a major decrease in α-terpinyl acetate and hexyl acetate was observed (Figure 3C). These compounds have been previously associated with orange lentils [37]. Ethenyl hexanoate has the highest negative VID in chickpeas and kidney beans; its formation is linked to the condensation of hexanoic acid and ethanol and is characterized by fruity, green, pleasant pineapple notes [38]. Since both hexanoic acid and ethanol were compounds selected with high negative VID, the (hydrothermal) inactivation of enzymes responsible for their evolution also indirectly decreased the production of ethenyl hexanoate. Likewise, the decrease of hexyl acetate in cowpeas is likely due to the limitation of their precursor through enzyme inactivation—in this case, 1-hexanol. Therefore, it appears that hydrothermal processing caused a decrease in these compounds through the precursor limitation via enzyme inactivation.

#### 2.5.2. Temperature-Related Reaction Pathways

The changes of furans, sulphur-containing compounds, benzopyrone, ketones and terpenes can be related to reactions associated with the high thermal load of the processing (such as the Maillard and temperature-related degradations). In the open system, 2-ethyl-furan and 2-pentyl-furan in kidney beans decreased as processing proceeded, likely due to evaporation into the environment. In contrast, an apparent increase in the amount of these volatile compounds was observed in the closed system (Figure 4A). Furan and their analogues are highly volatile and odour-active compounds and are also listed as a possible human carcinogen [39]. Hence, lower levels of furanic compounds would be desirable. Furanic compounds are commonly formed through multiple thermally-induced reaction pathways, including the Maillard reaction and the degradation of ascorbic acid, sugars, amino acids and/or unsaturated fatty acids [40]. In the case of legumes, precursors may include amino acids and oleic, linoleic and α-linolenic acids [41]. Furans may also be present in dried legume samples [37]. Thus, these highly volatile compounds may be formed during seed drying and subsequent boiling but are being leached into the cooking water and/or lost into the environment [39], accounting for the decrease in the opened system (Figure 4B). Indeed, Mishra, Tripathi, Gupta and Variyar [23] reported both the formation and disappearance of furans during their hydrothermal processing of red kidney beans. The present work demonstrated the importance of investigating both opened cooking and sealed cooking, as both systems have important and differing consequences for volatile formation.

In the closed system, sulphur-containing compounds, such as dimethyl sulfide and 2-acetylthiazole, were detected in the headspace of processed legumes (Table 3). Dimethyl sulfide has been described as possessing a cabbage-like odour and was reported in three types of hydrothermally processed kidney beans [23] and sterilised chickpeas [21]. The odour of 2-acetylthiazole has been described as roasty and popcorn-like [42] and is reported in serialized chickpeas [21]. Compared to the open system, where neither compound was not detected, Figure 4B illustrates an increasing trend in the closed system. A lack of detection in the opened system is likely due to loss to the environment, mitigated by processing in a sealed vessel. Sulphur-containing compounds can contribute to pungent, sulphurous odours, which may have flavour consequences in the prolonged processing and storage of canned beans. These compounds may have derived from (thermal) degradation of sulphur-containing amino acids, such as methionine and cysteine [43].

In the closed system, pyridine was found to increase majorly during the hydrothermal processing of chickpeas (Figure 4B). Pyridine has been reported to be associated with sterilised chickpeas [21]. It is a volatile heterocyclic compound related to benzene, with a nitrogen replacement on the ring. The formation of pyridine appears to have an initial lag phase up to *t_c_* = 30 min, followed by a significant increase. Pyridine may be formed from the interaction between the Maillard reaction intermediate products and other components [43]. Therefore, it is reasonable to assume that the lag/induction time is necessary for the Maillard reaction to proceed sufficiently. Furthermore, the current investigation is the first to detect pyridine during hydrothermal processing in both the cowpea and kidney bean.

In both the open and closed systems, 2H-benzopyran was found to increase in kidney beans significantly. Due to the retainment of volatile compounds in the closed system, maltol was also observed to increase in hydrothermally processed kidney beans. Maltol is a 4H-pyran, meaning that the saturated carbon is at position four of the ring; its odour has been described as sweet like caramel and cotton candy, and it is used as a flavour enhancer. Figure 4C illustrates 2H-benzopyran’s increasing abundance throughout processing. This agrees with Chigwedere, et al. [44], who also detected benzopyran in thermally treated kidney beans. Benzopyran forms from the fusion of a heterocyclic pyran ring and a benzene ring. It has been hypothesised as a thermal degradation product of the coloured compound found within kidney beans, likely from flavanones and isoflavones [44,45]. This is possible as kidney beans used in this study have a rich, dark and red-purple colour.

In the closed system processing, specific ketones were observed to increase, such as 6-methyl-5-hepten-2-one in cowpeas, 6-methyl-2-heptanone in chickpeas and 3-methyl-2-butanone in kidney beans (Table 3). Ma, Boye, Azarnia and Simpson [25] also observed an increase of 2-heptanone through the thermal treatment of various legumes (including kidney beans). This may be because of the elevated temperature catalysing the oxidation of primary and secondary alcohols and the degradation of carotenoids [10]. For example, 6-methyl-5-hepten-2-one is a volatile that is commonly associated with the thermal degradation of carotenoids [46]. The varying trends observed with ketones underline the importance of considering volatile analysis in both opened and closed systems.

It was observed that not all terpenes were similarly affected by boiling time. For example, in kidney beans, linalool decreased in both opened and closed systems. A similar trend was observed for cowpeas, with a loss of linalool, estragole, eucalyptol and carvone in the open system and a loss of d-limonene, cymene and eucalyptol in the closed system. This agrees with Mishra, Tripathi, Gupta and Variyar [23], who also reported a loss of terpene in cooked kidney beans. Moreover, in kidney beans, xylene decreased, a compound derived from lipids that is common to Phaseolus legume cultivars (Oomah et al., 2007). Thus, in addition to the possible loss to the environment, this can be attributed to thermally-induced degradation [47,48]. On the other hand, γ-terpinene was found to increase as a function of hydrothermal processing time in chickpeas. γ-terpinene has been reported as a volatile compound resulting from limonene degradation [45]. γ-terpinene has a sweet, citrus odour with tropical and lime nuances [49].

## 3. Materials and Methods

### 3.1. Sample Preparation and Storage

A batch of 7 kg commercial dried cowpeas (*Vigna unguiculata*), chickpeas (*Cicer arietinum*) and kidney beans (*Phaseolus vulgaris*) were purchased from a local market in Dunedin in August 2017. Deformed and damaged seeds were discarded. The seeds were then vacuum-packed in opaque aluminium bags and stored at 4 °C until processing. Analysis took place between late 2018 and early 2019.

### 3.2. Compositional Analyses of Uncooked Cowpeas, Chickpeas and Kidney Beans

Starch was estimated via the glucose oxidase/peroxidase assay [50]. Protein was estimated through total nitrogen via the Kjeldahl method [51]. Lipid was estimated by Soxtec, based on the Soxhlet method [33]. Moisture was estimated by the oven drying method based on AOAC Official Methods of Analysis method 930.04 [52].

### 3.3. Hydrothermal Processing of Cowpeas, Chickpeas and Kidney Beans

Hydrothermal processing of cowpeas, chickpeas and kidney beans was conducted in an open system and a closed system. For the opened system, the legumes (500 g) were soaked in distilled water at a seed-to-water ratio of 1:5 (*w/v*) for 16 h at 20 °C in an incubator (IL-11-4C, Lab Companion, MA, USA), after which the water was discarded. The legumes were aliquoted (~100 g) and transferred into six stainless-steel sieve cages to facilitate independent sampling at different time points (i.e., 15, 30, 45, 60, 90 and 120 min). Two stainless-steel boiling pots (10 L) and induction hot plates (2000-watt power, Micasa, Auckland, New Zealand) were used. The legumes were hydrothermally processed (~95 °C) in distilled water (1:20 (*w/v*) seed-to-water ratio). At each time, the specified sample was removed and immediately cooled to stop the treatment. Finally, legumes were aliquoted into aluminium parcels, snapped frozen in liquid nitrogen, and stored at −81 °C until analysis. Note that though 15 min may be insufficient to cook legumes, the duration was included to discern any trends in the volatile compounds at a short processing time.

For the closed system, legume seeds (7 g) were placed inside a Teflon-sealed glass test tube (2 mm thickness) and filled with distilled water at a seed-to-water ratio of 1:5 (*w/v*) and incubated for 16 h at 20 °C in an incubator (IL-11-4C, Lab Companion, MA, USA). The sealed test tubes were immediately processed by immersion into boiling water for varying durations (15, 30, 45, 60, 90 and 120 min) using the same induction hot plates as opened system cooking. The tube containing pre-soaked legumes was not hydrothermally processed to represent time of 0 min (i.e., control of the cooking experiment). At each sampling time, test tubes with cooked samples inside were removed and immersed in an ice-water bath to stop the heat treatment. The processed legume was then homogenised at 3000 rpm (ULTRA-TURRAX, Guangzhou, China) for 20 s in a 4 °C walk-in cold room, with the processing water to produce legume slurry. This was done to capture the volatile compounds which may have migrated into the processing water. Finally, the homogenised legume slurry was aliquoted into test tubes and stored at −20 °C until analysis.

### 3.4. Determination of Fatty Acids in Cowpeas, Chickpeas and Kidney Beans Using Chromatography Flame Ionisation Detection

Legume lipid was extracted with the Soxhlet method and converted to fatty acid methyl esters (FAME), and then detected using gas chromatography flame ionisation detection (GC-FID) according to the AOAC method 963.22 [53], with modifications. To facilitate the Soxhlet solvent lipid extraction, samples were thawed (4 °C for 12 h) then dried overnight at 50 °C (7100, Contherm, Hutt City, New Zealand) for 16 h to ensure a constant weight. The dried legumes were ground into flour in a mortar and pestle at an ambient temperature (~20 °C). The resulting flour was sieved to pass through an 850 µm mesh. Flour retaining between a 450 and 850 µm mesh size was used for lipid analysis to extract lipids consistently.

The remaining procedure for the fatty acid profile analysis is described in Khrisanapant, Kebede, Leong and Oey [37] for the extraction of lipid, lipid purification, lipid esterification and fatty acid profiling using GC-FID. The fatty acid profile of legumes was analysed in three independent replicates.

For fatty acid data analysis, chromatograms obtained from GC-FID were analysed with GC ChemStation (Build 4.01, Agilent Technologies, Santa Clara, CA, USA), and individual peaks were manually identified by matching the retention time with commercial standards (FAMQ-005, AccuStandards, New Haven, CT, USA; Appendix A). Following the manual peak alignment and the removal of interfering background compounds, the proportion of signal abundance of each fatty acid was calculated in a percentage abundance of total signal abundance. Significant differences in the fatty acid profile between legumes were tested using ANOVA with Tukey’s post hoc significance testing (*p* < 0.05).

### 3.5. Determination of Volatile Compounds Using Headspace Solid-Phase Micro-Extraction Gas Chromatography-Mass Spectrometry

Headspace solid-phase micro-extraction gas chromatography-mass spectrometry (HS-SPME-GC-MS) was conducted according to the work of Kebede, Grauwet, Tabilo-Munizaga, Palmers, Vervoort, Hendrickx and Van Loey [54] and Liu, Grauwet, Kebede, Van Loey, Liao and Hendrickx [55] with modifications.

Each legume sample was gently thawed overnight (4 °C). For the opened system, thawed legumes were disintegrated in a mortar and pestle, weighed to reach 0.6 g DW and placed into individual 20 mL glass vials, which were topped up with water to reach 3.0 g (moisture contents were previously determined). For the closed system, the legume slurry was directly weighed (3.0 g) into glass vials. In both instances, the total solid was maintained at 0.6 g. After that, 5 mL of saturated sodium chloride solution (360 g/L) was added to increase the solution’s ionic strength and drive the legume volatiles into the headspace. The vials were securely sealed with PTFE/silicone septum and crimp caps.

Using the Gerstel MPS Maestro autosampler (Gerstel, Linthicum Heights, MD, USA), each sample was incubated at 40 °C for 5 min, with agitation at 250 rpm. After that, the volatile compounds were extracted using HS-SPME (four replicates). A preconditioned (according to the manufacturer’s instructions) SPME fibre with a 30/50 μm divinylbenzene/carboxen/polydimethylsiloxane (DVB/CAR/PDMS) sorptive coating (Stableflex, Supelco, Bellefonte, PA, USA) was used to extract a wide range of volatile compounds from the headspace of the sample vial for 30 min at 40 °C.

For the GC-MS analysis (Agilent 6890N, Agilent Technologies, Santa Clara, CA, USA), the extracted volatiles were thermally desorbed in the injection port at 230 °C for 2 min, then injected in splitless mode onto a ZB-Wax capillary column (30 m × 0.25 mm × 0.25 μm; Agilent Technologies, Santa Clara, CA, USA) for separation with helium as the carrier gas at 1.5 mL/min. The GC oven was maintained at 50 °C for 5 min before the temperature was ramped up to 210 °C at 5 °C/min, after which it was again ramped to 240 °C at the rate of 10 °C/min for a total GC-MS run time of 37 min. For the MS, the quadrupole was set at 70 eV, and the ion sources were 150 °C and 230 °C, respectively, with a mass-to-charge ratio scanning range of 30–300 *m/z*. The SPME fibre was regenerated according to the manufacturer’s instructions.

### 3.6. Pre-Processing of Headspace Volatile Chromatograms

Data pre-processing procedures were conducted according to Arcena, et al. [56]. Briefly, after obtaining the total ion chromatogram for each sample treatment (Appendix A), the GC-MS data file was pre-processed with the automated mass spectral deconvolution and identification system (AMDIS; version 2.72, build 140.24, Agilent Technologies, Santa Clara, CA, USA) to deconvolute overlapping peaks and filter interferences. After that, the deconvoluted spectrum was further processed with the Mass Profiler Professional (MPP; version 14.9.1, build 1316, Agilent Technologies, Santa Clara, CA, USA) to filter non-reproducible peaks and align them. This was followed by a manual checking and tentative identification of each volatile compound using the following three criteria to improve the confidence of identification: (i) match and reverse match with the NIST spectra library of no less than 90%; (ii) comparison of experimental retention index with RI, according to the literature; and (iii) matching retention time and spectra with authentic standards from different chemical groups of detected volatiles.

### 3.7. Data Analysis

Chemometrics was applied using principal component analysis (PCA), followed by partial least square regression (PLS-R), utilising Solo software (Version 8.6, Eigenvector Research, Manson, WA, USA). Firstly, PCA was used as an unsupervised technique to explore the trends and detect outliers. Secondly, PLS-R was used as a supervised regression technique to further investigate the evolution of the volatile organic compounds (the X-variables) as a function of the hydrothermal processing time (*t_c_*; Y-variables). Bi-plot was generated as a visual representation of the classification (OriginPro, OriginLab, Northampton, MA, USA).

Volatile compounds that changed due to hydrothermal processing time were selected by calculating the variable identification (VID) coefficients [54,56]. VID values represent the correlation coefficients between X-variables (volatile compounds) and predicted Y-variables (hydrothermal processing time). Volatile compounds with an absolute threshold value of >|0.600| were considered selected. Compounds chosen at a |0.600| threshold were confirmed with significance (*p* < 0.05) testing.

## 4. Conclusions

This work successfully monitored the volatile changes of legumes during hydrothermal processing. The chemometrics and feature selection methods successfully discerned key patterns, marker compounds and associated reaction pathways. Alcohol, aldehyde, ketone, terpene, acid, ester and lactone volatile compounds showed decreasing trends during hydrothermal processing. This can be due to the: i) inhibition of volatile formations through enzyme inactivation; ii) loss of pre-existing volatile compounds through process-induced degradation; and iii) loss into the environment (mainly in the opened system). Most of these compounds (especially the aldehydes and alcohols) possess undesirable odours. This work showed that between 15 to 30 min of hydrothermal processing (~95 °C) is needed to significantly decrease the amount of these compounds that are commonly associated with undesirable odour. Other volatiles, such as furanic compounds, pyrans and sulphur compounds, showed an increasing trend as a function of hydrothermal processing time. The potential reaction pathways involved in the increased formation of these compounds can be attributed to the thermal-induced degradation of constituents and the Maillard reactions (including Strecker degradation). Most of these highly volatile compounds seem to be retained in the closed system compared to the open system. The present work provides valuable insight into the volatile changes of legumes in two approaches: boiling in an open system to approximate standard household cooking practices and boiling in a closed system to represent the way legumes are being processed industrially. It gives consumers and the food industry valuable information on selecting a processing intensity to maximise legume utilisation. In the future, sensory analysis can be conducted to support the observations of the instrumental analysis. Moreover, understanding the effect of other non-hydrothermal processed methods, such as smoking and pickling, on the legume’s volatile profile could be considered in forthcoming studies.

## Figures and Tables

**Figure 1 molecules-27-08204-f001:**
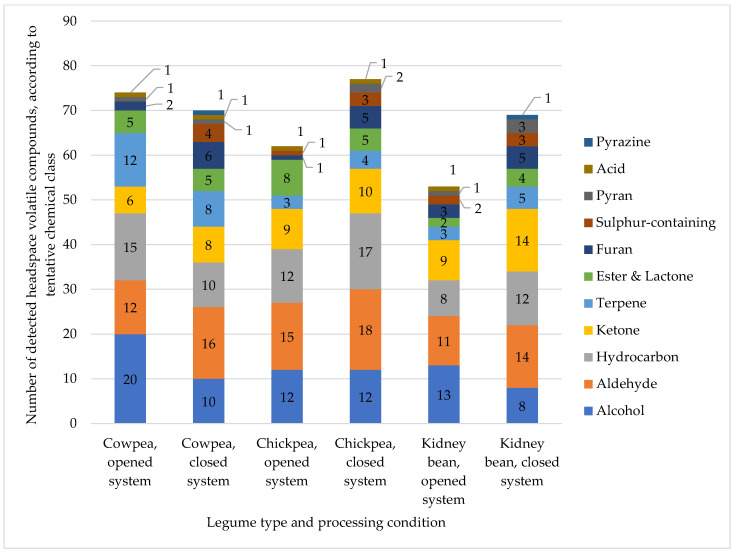
Comparison of number of detected headspace volatile compounds between the opened and closed system of processing, with tentative chemical classes identified.

**Figure 2 molecules-27-08204-f002:**
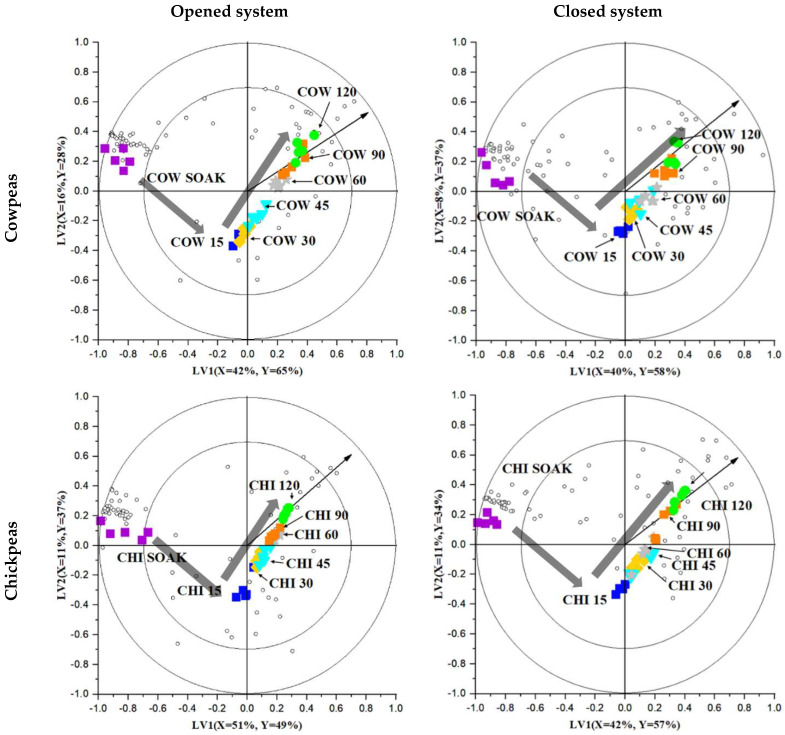
A bi-plot based on partial least square regression (PLS-R) comparing the volatile profiles among treatments of three types of legumes as a function of the hydrothermal processing time (0 (labelled SOAK on the figure), 15, 30, 45, 60, 90, 120 min). The thick transparent grey arrows are added to indicate the change from soaked to processed samples. The percentages of X- and Y-variances explained by each latent variable (LV) are specified on each respective axis.

**Figure 3 molecules-27-08204-f003:**
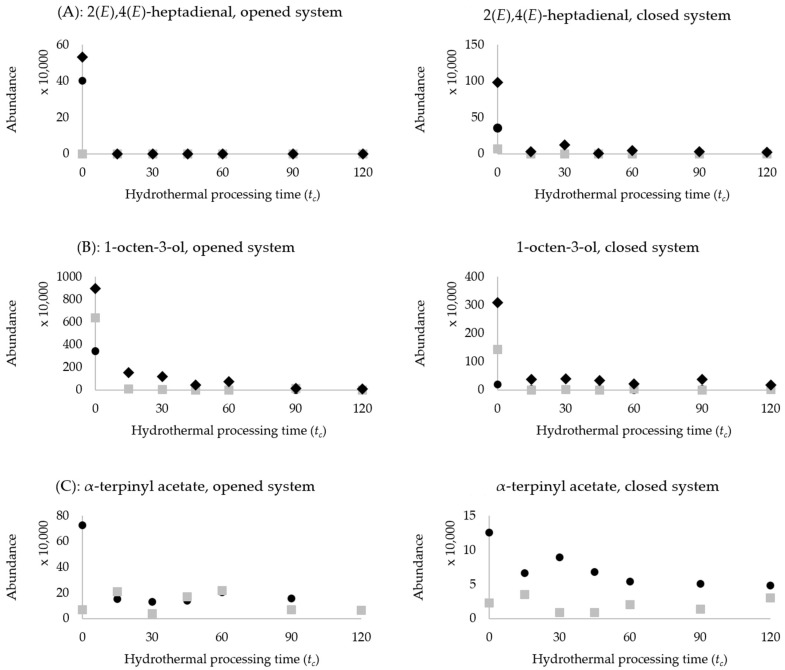
Scatter plots comparing enzyme-related changes in legumes’ headspace volatile compounds in opened versus closed system of hydrothermal processing. Legend: 
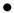
 represents cowpeas; 
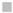
 represents chickpeas; 
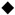
 represents kidney beans. (**A**): 2(*E*),4(*E*)-heptadienal, opened and closed system, (**B**): 1-octen-3-ol, opened and closed system, (**C**): α-terpinyl acetate, opened and closed system.

**Figure 4 molecules-27-08204-f004:**
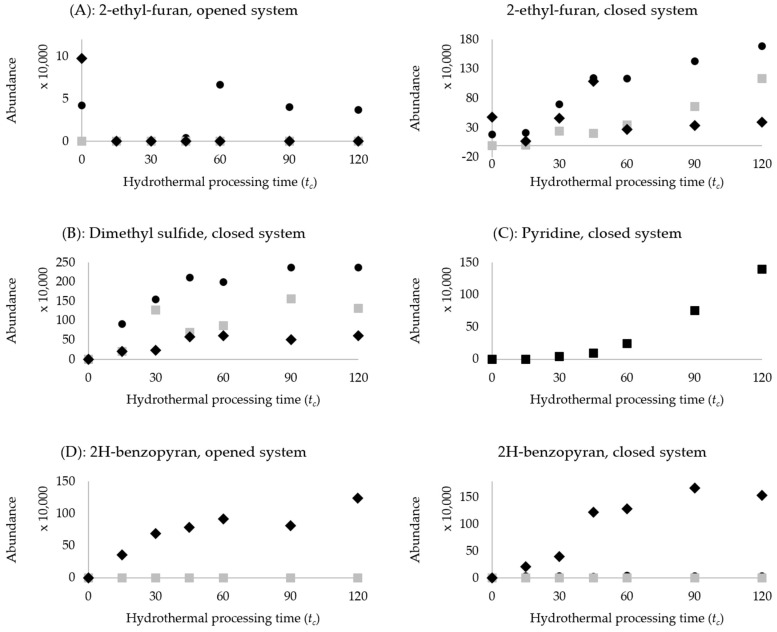
Scatter plots comparing thermal-related changes in legumes’ headspace volatile compounds in opened versus closed system of hydrothermal processing. Legend: 
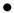
 represents cowpeas; 
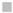
 represents chickpeas; 
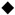
 represents kidney beans. (**A**): 2-ethyl-furan, opened and closed system, (**B**): Dimethyl sulfide, closed system, (**C**): Pyridine, closed system, (**D**): 2H-benzopyran, opened and closed system.

**Table 1 molecules-27-08204-t001:** Lipid yield and relative fatty acid abundance (g/100 g lipid extract) of (cooked) cowpeas, chickpeas and kidney beans as affected by hydrothermal processing at varying durations.

	*t_c_* (min)	Lipid Yield (g/100 g DW)	C16:0 (g/100 g Lipid)	C18:0 (g/100 g Lipid)	C18:1 (g/100 g Lipid)	C18:2 (g/100 g Lipid)	C18:3 (g/100 g Lipid)	SFAs (g/100 g Lipid)	MUFAs (g/100 g Lipid)	PUFAs (g/100 g Lipid)	ω-6/ω-3 Ratio
Cowpea	0	3.31 ± 0.18 ^c^	27.43 ± 0.19 ^a^	4.41 ± 0.24 ^a^	5.48 ± 0.18 ^a^	36.51 ± 0.23 ^a^	24.99 ± 0.21 ^ab^	31.84 ± 0.15 ^b^	5.48 ± 0.18 ^a^	61.50 ± 0.17 ^a^	1.46 ± 0.02 ^a^
	15	2.81 ± 0.06 ^b^	26.95 ± 0.65 ^ab^	4.21 ± 0.12 ^a^	6.06 ± 0.11 ^a^	36.57 ± 0.41 ^ab^	24.72 ± 0.29 ^a^	31.17 ± 0.74 ^ab^	6.06 ± 0.11 ^a^	61.29 ± 0.65 ^a^	1.48 ± 0.01 ^abc^
	30	2.54 ± 0.10 ^ab^	26.16 ± 0.39 ^b^	4.04 ± 0.05 ^a^	5.85 ± 0.30 ^a^	36.67 ± 0.34 ^ab^	24.94 ± 0.23 ^a^	30.21 ± 0.37 ^a^	5.85 ± 0.30 ^a^	61.61 ± 0.55 ^ab^	1.47 ± 0.01 ^ab^
	45	2.43 ± 0.08 ^a^	27.04 ± 0.66 ^ab^	4.03 ± 0.18 ^a^	6.18 ± 0.71 ^a^	37.79 ± 0.42 ^c^	24.73 ± 0.15 ^a^	31.07 ± 0.79 ^ab^	6.18 ± 0.71 ^a^	62.51 ± 0.53 ^abc^	1.53 ± 0.01 ^d^
	60	2.39 ± 0.05 ^a^	27.04 ± 0.06 ^ab^	3.99 ± 0.22 ^a^	6.01 ± 0.26 ^a^	37.41 ± 0.13 ^abc^	25.57 ± 0.29 ^b^	31.03 ± 0.25 ^ab^	6.01 ± 0.26 ^a^	62.97 ± 0.38 ^c^	1.46 ± 0.01 ^a^
	90	2.41 ± 0.22 ^a^	26.84 ± 0.39 ^ab^	4.05 ± 0.15 ^a^	6.10 ± 0.17 ^a^	37.49 ± 0.68 ^bc^	24.92 ± 0.37 ^a^	30.89 ± 0.45 ^ab^	6.10 ± 0.17 ^a^	62.41 ± 0.96 ^abc^	1.50 ± 0.02 ^bcd^
	120	2.66 ± 0.2 ^ab^	26.60 ± 0.40 ^ab^	4.36 ± 0.38 ^a^	5.65 ± 0.27 ^a^	37.90 ± 0.39 ^c^	24.99 ± 0.30 ^ab^	30.95 ± 0.19 ^ab^	5.65 ± 0.27 ^a^	62.90 ± 0.44 ^bc^	1.52 ± 0.03 ^cd^
Chickpea	0	8.17 ± 0.25 ^a^	10.30 ± 0.05 ^a^	1.98 ± 0.03 ^d^	37.47 ± 0.20 ^a^	46.32 ± 0.20 _e_	2.12 ± 0.02 ^a^	12.28 ± 0.07 ^b^	37.47 ± 0.2 ^a^	48.45 ± 0.20 ^e^	21.81 ± 0.18 ^d^
	15	8.73 ± 1.44 ^a^	10.28 ± 0.03 ^a^	1.74 ± 0.04 ^bc^	38.31 ± 0.10 ^b^	45.65 ± 0.13 ^d^	2.14 ± 0.01 ^ab^	12.03 ± 0.02 ^a^	38.31 ± 0.1 ^b^	47.78 ± 0.13 ^d^	21.35 ± 0.11 ^cd^
	30	8.23 ± 0.96 ^a^	10.29 ± 0.05 ^a^	1.74 ± 0.03 ^bc^	38.88 ± 0.04 ^cd^	45.07 ± 0.03 _c_	2.12 ± 0.03 ^a^	12.04 ± 0.04 ^a^	38.88 ± 0.04 ^cd^	47.18 ± 0.02 ^bc^	21.30 ± 0.33 ^cd^
	45	8.48 ± 0.32 ^a^	10.71 ± 0.12 ^c^	1.82 ± 0.050 ^c^	39.31 ± 0.14 ^d^	44.13 ± 0.30 _a_	2.13 ± 0.05 ^a^	12.54 ± 0.17 ^c^	39.31 ± 0.14 ^d^	46.26 ± 0.29 ^a^	20.75 ± 0.56 ^bc^
	60	8.33 ± 0.70 ^a^	10.42 ± 0.05 ^ab^	1.70 ± 0.040 ^ab^	39.30 ± 0.19 ^d^	44.49 ± 0.16 ^ab^	2.27 ± 0.05 ^c^	12.11 ± 0.03 ^ab^	39.30 ± 0.19 ^d^	46.77 ± 0.20 ^ab^	19.59 ± 0.36 ^a^
	90	7.93 ± 1.15 ^a^	10.43 ± 0.02 ^ab^	1.66 ± 0.02 ^ab^	38.82 ± 0.17 ^cd^	44.93 ± 0.10 ^bc^	2.25 ± 0.05 ^bc^	12.09 ± 0.03 ^ab^	38.82 ± 0.17 ^cd^	47.19 ± 0.15 ^bc^	19.96 ± 0.43 ^ab^
	120	7.93 ± 0.27 ^a^	10.49 ± 0.04 ^b^	1.64 ± 0.02 ^a^	38.43 ± 0.29 ^bc^	45.31 ± 0.27 ^cd^	2.23 ± 0.05 ^abc^	12.13 ± 0.05 ^ab^	38.43 ± 0.29 ^bc^	47.54 ± 0.32 ^cd^	20.37 ± 0.36 ^abc^
Kidney bean	0	3.36 ± 0.19 ^a^	18.72 ± 0.75 ^a^	3.14 ± 0.53 ^a^	14.47 ± 3.41 ^a^	29.25 ± 2.00 ^a^	33.93 ± 1.66 ^a^	21.86 ± 0.6 ^a^	14.47 ± 3.41 ^a^	63.19 ± 3.59 ^ab^	0.86 ± 0.03 ^a^
	15	3.10 ± 0.32 ^a^	18.59 ± 3.19 ^a^	3.43 ± 0.79 ^a^	12.49 ± 0.90 ^a^	28.11 ± 2.28 _a_	36.85 ± 2.26 ^a^	22.03 ± 3.88 ^a^	12.49 ± 0.9 ^a^	64.96 ± 4.50 ^ab^	0.76 ± 0.02 ^a^
	30	3.00 ± 0.16 ^a^	18.66 ± 1.74 ^a^	2.98 ± 0.67 ^a^	15.94 ± 4.53 ^a^	27.01 ± 1.92 ^a^	34.23 ± 2.87 ^a^	21.64 ± 2.40 ^a^	15.94 ± 4.53 ^a^	61.24 ± 4.79 ^ab^	0.79 ± 0.01 ^a^
	45	3.32 ± 0.17 ^a^	18.30 ± 0.52 ^a^	2.78 ± 0.37 ^a^	14.12 ± 0.74 ^a^	28.54 ± 0.31 ^a^	35.61 ± 0.52 ^a^	21.08 ± 0.63 ^a^	14.12 ± 0.74 ^a^	64.15 ± 0.65 ^ab^	0.80 ± 0.01 ^a^
	60	3.71 ± 0.43 ^a^	20.07 ± 1.03 ^a^	3.73 ± 1.12 ^a^	17.46 ± 3.52 ^a^	24.94 ± 3.63 ^a^	32.95 ± 2.40 ^a^	23.80 ± 1.82 ^a^	17.46 ± 3.52 ^a^	57.89 ± 4.60 ^a^	0.76 ± 0.11 ^a^
	90	3.30 ± 0.48 ^a^	17.42 ± 0.13 _a_	2.25 ± 0.40 ^a^	12.40 ± 0.41 ^a^	30.25 ± 1.17 ^a^	37.13 ± 1.17 ^a^	19.67 ± 0.45 ^a^	12.40 ± 0.41 ^a^	67.38 ± 0.73 ^b^	0.82 ± 0.05 ^a^
	120	3.07 ± 0.36 ^a^	17.28 ± 0.72 ^a^	4.11 ± 1.01 ^a^	13.65 ± 2.43 ^a^	28.84 ± 1.81 ^a^	35.69 ± 2.01 ^a^	21.40 ± 1.65 ^a^	13.65 ± 2.43 ^a^	64.53 ± 3.72 ^ab^	0.81 ± 0.02 ^a^

*t_c_* represents hydrothermal processing time (min). *t_c_* = 0 represented soaked beans immediately before hydrothermal processing. C16:0 = palmitic acid; C18:0 = stearic acid; C18:1 = oleic acid; C18:2 = linoleic acid; C18:3 = α-linolenic acid; SFAs = saturated fatty acids; MUFAs = monounsaturated fatty acids; PUFAs = polyunsaturated fatty acid; ω-6/ω-3 = ratio of omega-6 to omega-3 fatty acids. Values expressed as mean ± standard deviation (n = 3). Means with different superscripts in the same column indicate a significant difference (*p* < 0.05). The fatty acid proportions were analysed by fatty acid methyl ester gas chromatography coupled with a flame ionisation detector (FAME-GC-FID).

**Table 2 molecules-27-08204-t002:** Discriminant volatile compounds changing as a function of hydrothermal processing time (*t_c_*) in opened (home) processing of cowpeas, chickpeas and kidney beans as tentatively identified by NIST spectra, matching with the literature retention index (RI), and authentic standards.

VID Coefficient	Tentative Identity	RI_observed_	RI_literature_
Cowpea			
Positive VID			
Aldehyde (1)			
0.602	Benzaldehyde	1554	1543
Hydrocarbon (1)			
0.620	Branched hydrocarbon	988	N/A
Terpene (1)			
0.763	α-Muurolene	1726	1734
Negative VID			
Alcohol (14)			
−0.754	1-Heptanol	1467	1462
−0.711	1-Pentanol	1257	1255
−0.706	1-Nonanal	1419	1412
−0.705	1-Hexanol	1365	1357
−0.704	2-Ethyl-1-hexanol	1499	1490
−0.700	1-Penten-3-ol	1156	1174
−0.699	3(*Z*)-Hexen-1-ol	1400	1381
−0.695	1-Octanol	1562	1549
−0.689	1-Octen-3-ol	1461	1461
−0.675	Ethanol	912	939
−0.666	3-Octanol	1405	N/A
−0.658	2(*Z*)-Penten-1-ol	1333	1318
−0.652	1-Heptanol	1467	1462
−0.650	3-Methyl-1-butanol	1210	1214
Aldehyde (8)			
−0.745	2(*E*)-Nonenal	1547	1550
−0.741	5-Ethylcyclopent-1-enecarboxaldehyde	1451	1416
−0.712	Nonanal	1419	N/A
−0.699	2(*E*)-Hexenal	1236	1238
−0.680	Pentadecanal	1945	2024
−0.649	Hexanal	1076	1078
−0.640	2(*E*),4(*E*)-Heptadienal	1491	1471
−0.623	2-Butenal	1603	N/A
Ester and Lactone (2)			
−0.693	α-Terpinyl acetate	1698	1704
−0.600	Hexyl acetate	1289	1259
Ketone (2)			
−0.693	3-Hydroxy-2-butanone (Acetoin)	1309	1295
−0.665	3-Octen-2-one	1434	1414
Terpene (5)			
−0.727	Linalool	1552	1551
−0.707	Estragole	1677	1687
−0.649	Terpinen-4-ol	1615	1611
−0.634	Eucalyptol	1225	1213
−0.622	Carvone	1736	1737
Chickpea			
Positive VID			
Hydrocarbon (1)			
0.696	Branched hydrocarbon	1010	N/A
Terpene (1)			
0.646	γ-Terpinene	1267	1255
Negative VID			
Acid (1)			
−0.687	Hexanoic acid	1816	1865
Alcohol (11)			
−0.895	Ethanol	912	939
−0.815	1-Octanol	1562	1549
−0.803	1-Hexanol	1365	1357
−0.801	1-Heptanol	1467	1462
−0.801	1-Penten-3-ol	1156	1174
−0.801	1-Pentanol	1257	1255
−0.799	1-Octen-3-ol	1461	1461
−0.791	3(*Z*)-Hexen-1-ol	1400	1381
−0.780	2(*Z*)-Penten-1-ol	1333	1318
−0.716	4-Ethylcyclohexanol	1520	N/A
−0.678	3-Hydroxy-2-butanone (Acetoin)	1309	1295
Aldehyde (14)			
−0.820	Nonanal	1419	1412
−0.800	2(*E*)-Decenal	1652	1655
−0.795	2(*Z*)-Heptenal	1352	1349
−0.793	5-Ethylcyclopent-1-enecarboxaldehyde	1451	1416
−0.786	Heptanal	1196	1202
−0.782	2(*E*)-Octenal	1458	1466
−0.776	2(*E*),4(*E*)-Heptadienal	1491	1503
−0.774	Pentanal	963	985
−0.758	Hexanal	1076	1068
−0.740	Octanal	1311	1307
−0.726	2-Undecenal	1741	N/A
−0.700	2-Ethyl-4-pentenal	1324	N/A
−0.695	2,4-Decadienal	1790	1767
−0.667	3-Methyl-1-butanal	904	924
Ester and Lactone (3)			
−0.792	Hexanoic acid, ethenyl ester	1667	N/A
−0.752	Hexahydro-2,5-methano-2H-furo [3,2-b]pyran	1263	N/A
−0.660	Dihydro-5-pentyl-2(3H)-furanone	1957	2005
Furan (1)			
−0.732	2-Pentyl-furan	1247	1224
Hydrocarbon (1)			
−0.681	2-Methoxy-2-propenyl-benzene	1805	N/A
−0.656	1-Ethyl-1-methyl-cyclopentane	1430	N/A
Ketone (4)			
−0.709	2,5-Octanedione	1344	N/A
−0.695	3-Octen-2-one	1434	1414
−0.692	2-Octanone	1306	1323
−0.686	6-Methyl-3-heptanone	1273	N/A
Kidney bean			
Positive VID			
Pyran (1)			
0.740	2H-1-benzopyran	1553	N/A
Negative VID			
Acid (1)			
−0.712	Hexanoic acid	1816	1833
Alcohol (10)			
−0.863	1-Octen-3-ol	1461	1461
−0.819	1-Penten-3-ol	1156	1174
−0.817	1-Hexanol	1365	1357
−0.815	3(*Z*)-Hexen-1-ol	1400	1381
−0.794	1-Heptanol	1467	1462
−0.779	3-Methyl-1-butanol	1209	1214
−0.765	1-Butoxy-2-propanol	1358	N/A
−0.734	2(*Z*)-Penten-1-ol	1333	N/A
−0.728	1-Pentanol	1257	1255
−0.721	Ethanol	912	939
Aldehyde (9)			
−0.815	2(*E*)-Hexenal	1236	1238
−0.790	Hexanal	1076	1078
−0.786	2(*E*),4(*E*)-Heptadienal	1520	1471
−0.786	2(*E*)-Nonenal	1547	1550
−0.786	2(*E*)-Pentenal	1134	1146
−0.784	2(*Z*)-Heptenal	1352	1349
−0.779	3-Methyl-1-butanal	904	924
−0.703	2(*E*)-Octenal	1458	1466
−0.663	Pentanal	963	985
Ester and Lactone (1)			
−0.747	Hexanoic acid, ethenyl ester	1667	N/A
Furan (1)			
−0.718	2-Ethyl-furan	937	945
Hydrocarbon (3)			
−0.815	Naphthalene	1749	1765
−0.682	Toluene	1031	1022
Ketone (3)			
−0.803	2,3-Pentanedione	1046	1050
−0.699	3(*E*),5(*E*)-octadien-2-one	1540	1531
−0.678	3-Hydroxy-2-butanone (Acetoin)	1309	1295
Terpene (1)			
−0.664	Linalool	1552	1551

Each volatile compound’s retention index was obtained from the NIST library of published studies. N/A—These volatile compounds, tentatively identified by matching with NIST spectra, did not have available the literature retention index information.

**Table 3 molecules-27-08204-t003:** Discriminant volatile compounds changing during closed system hydrothermal processing of cowpeas, chickpeas and kidney beans as tentatively identified by NIST spectra, matching with the literature retention index, and authentic standards.

VID Coefficient	Tentative Identity	RI_observed_	RI_literature_
Cowpea			
Positive VID			
Sulphur-containing (1)			
0.948	Dimethyl sulfide	802	777
Furan (2)			
0.869	2-Ethyl-furan	929	945
0.603	2-Methyl-furan	861	888
Ketone (1)			
0.628	6-Methyl-5-hepten-2-one	1355	1319
Terpene (1)			
0.600	γ-Terpinene	1260	1255
Negative VID			
Acid (1)			
−0.611	Hexanoic acid	1811	1865
Alcohol (5)			
−0.742	1-Hexanol	1364	1357
−0.738	3(*Z*)-Hexen-1-ol	1398	1381
−0.677	2-Penten-1-ol	1328	1321
−0.652	1-Nonanal	1415	1412
−0.618	1-Penten-3-ol	1149	1174
Hydrocarbon (5)			
−0.720	Decane	972	N/A
−0.675	Branched hydrocarbon	850	N/A
−0.662	Branched hydrocarbon	1116	N/A
−0.628	Toluene	1023	1014
−0.604	Branched hydrocarbon	1303	N/A
Aldehyde (9)			
−0.779	2(*E*)-Hexenal	1229	1238
−0.771	2(*E*)-Nonenal	1553	1550
−0.748	2(*E*),4(*E*)-Nonadienal	1700	1664
−0.748	Hexanal	1069	1078
−0.745	2(*E*)-Octenal	1453	1400
−0.736	2(*E*),4(*E*)-Heptadienal	1485	1471
−0.691	5-Ethylcyclopent-1-enecarboxaldehyde	1445	1416
−0.657	Heptanal	1189	1176
−0.632	Pentadecanal	1946	2024
Ester and Lactone (3)			
−0.742	Hexyl acetate	1283	1259
−0.737	α-Terpinyl acetate	1695	1704
−0.658	Ethyl 3-methylbutanoate	1049	1053
Terpene (3)			
−0.720	D-Limonene	1208	1190
−0.719	p-Cymene	1288	1280
−0.631	Eucalyptol	1219	1179
Furan (1)			
−0.490	2-Pentyl-furan	1240	1224
Ketone (1)			
−0.729	3,5-Dimethyl-2-octanone	1053	N/A
Sulphur−containing (1)			
−0.674	Hydroxyethyl methyl sulfide	1545	1537
Chickpea			
Positive VID			
Aldehyde (1)			
0.844	Benzaldehyde	1547	1506
Furan (1)			
0.832	2-Ethyl-furan	929	945
Ketone (2)			
0.739	6-Methyl-2-heptanone	1248	1319
0.725	2-Propanone	829	832
Pyridine (1)			
0.763	Pyridine	1185	1213
Sulphur−containing (2)			
0.774	Dimethyl sulfide	803	777
0.766	2-Acetylthiazole	1654	1650
Negative VID			
Acid (1)			
−0.646	Hexanoic acid	1811	1865
Alcohol (8)			
−0.754	1-Heptanol	1466	1462
−0.717	1-Octen-3-ol	1458	1461
−0.699	1-Hexanol	1364	1357
−0.662	1-Octanol	1562	1549
−0.656	2(*E*)-Hepten-1-ol	1518	1517
−0.639	2(*E*)-Octen-1-ol	1610	1580
−0.635	1-Nonanol	1652	1658
−0.625	1-Penten-3-ol	1149	1174
Ketone (1)			
−0.648	2-Octanone	1300	1323
Aldehyde (14)			
−0.745	2(*E*)-Nonenal	1554	1550
−0.741	5-Ethylcyclopent-1-enecarboxaldehyde	1445	1416
−0.735	2(*E*)-Decenal	1650	1655
−0.727	4-Oxononanal	1802	N/A
−0.715	Hexanal	1069	1078
−0.701	2(*E*),4(*E*)-Decadienal	1788	1770
−0.668	2(*E*)-Octenal	1453	1400
−0.666	Heptanal	1189	1176
−0.664	3-Methyl-butanal	898	934
−0.654	2(*E*),4(*E*)-Nonadienal	1700	1664
−0.653	2(*Z*)-Heptenal	1346	1349
−0.646	2-Undecenal	1740	1740
−0.629	Nonanal	1415	1412
−0.602	Octanal	1305	1307
Ester and Lactone (2)			
−0.740	Dihydro-5-pentyl-2(3h)-furanone	1956	2005
−0.678	Hexanoic acid, ethenyl ester	1663	N/A
Furan (1)			
−0.701	2-Pentyl-furan	1240	1224
Hydrocarbon (3)			
−0.657	1-Ethyl-1-methyl-cyclopentane	1425	N/A
−0.656	2-Methyl-6-propylphenol	1820	N/A
−0.644	Branched hydrocarbon	983	N/A
Pyran (1)			
−0.732	Hexahydro-2,5-methano-2H-furo [3,2-b]pyran	1257	N/A
Terpene (1)			
−0.648	p-Xylene	1141	1194
Kidney bean			
Positive VID			
Ketone (1)			
0.606	3-Methyl-2-butanone	954	949
Pyran (1)			
0.789	2H-1-Benzopyran	1549	N/A
Sulphur−containing (1)			
0.612	Dimethyl sulfide	802	777
Negative VID			
Alcohol (5)			
−0.746	3(*Z*)-Hexen-1-ol	1398	1381
−0.718	1-Hexanol	1364	1357
−0.713	1-Octen-3-ol	1458	1461
−0.705	1-Nonanol	1652	1658
−0.623	1-Octanol	1562	1549
Aldehyde (7)			
−0.686	2(*E*),4(*E*)-Heptadienal	1486	1471
−0.685	2(*E*)-Pentenal	1127	1146
−0.684	2(*E*)-Hexenal	1230	1238
−0.661	2(*E*)-Octenal	1453	1400
−0.655	Heptanal	1190	1176
−0.653	2(*E*)-Nonenal	1553	1550
−0.618	Hexanal	1069	1078
Hydrocarbon (1)			
−0.741	Branched hydrocarbon	1445	N/A
Ketone (2)			
−0.776	3,5-Octadien-2-one	1536	1531
−0.668	2-Octanone	1300	1323

VID stands for variable identification. N/A—These volatile compounds tentatively identified by matching with NIST spectra did not have available the literature retention index information.

## Data Availability

The data presented in this study are available in Appendix A.

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
