# Peer review of "Effects of Hydrothermal Processing on Volatile and Fatty Acids Profile of Cowpeas (*Vigna unguiculata*), Chickpeas (*Cicer arietinum*) and Kidney Beans (*Phaseolus vulgaris*)"

_molecules, 2022, doi:10.3390/molecules27238204_

Round 1

Reviewer 1 Report

This work studied the effects of hydrothermal processing on cow- peas, chickpeas, and kidney beans, with an approximation of home cooking (referred to as opened system) and an approximation of industrial cooking (referred to as closed system) of different processing time. In a detailed experiment, the impact of the process was discussed on the fatty acid and volatile profiles, and the evolution of an increased number of volatile compounds during processing was monitored by GC-MS-based fingerprinting method. Finally, chemometrics was adopted to help understand the key patterns and identify discriminant compounds. Overall, this is a complete study with solid data and good logic. However, some concerns are needed to be addressed. Therefore, I recommend a major revision prior to publication.

1.      What is the purpose of this work? It is not that clear. Please clarify it in the abstract.

2.      It is mentioned that legumes are not as fully utilized (partly) due to volatile compounds contributing to their undesirable odour. So, how do you define the undesirable odour? Is it a smell undesirable to a vast of people or to a small group ones?

3.      Actually, the factors involved in cooking time of beans are diverse and complex, why do the authors choose 95 °C in the experiments?

4.      Do other factors like environment, post-harvest storage and preprocessing influence the cooking time?

5.      It seems that this work aims to give consumers and food industry valuable information on selecting processing intensity to maximise legume utilization. Why only the hydrothermally processed method was discussed rather than other food processing methods like smoking or picking?

6.      How do you evaluate the odour? By sensory system or just by human nose? How do the authors judge the desirable level of odour?

Author Response

Dear, 

The authors thank the reviewer for providing valuable feedback in this peer review process. Please find each reviewer's point of comment and the author's explanation below.

Best regards, Biniam Kebede

This work studied the effects of hydrothermal processing on cow- peas, chickpeas, and kidney beans, with an approximation of home cooking (referred to as opened system) and an approximation of industrial cooking (referred to as closed system) of different processing time. In a detailed experiment, the impact of the process was discussed on the fatty acid and volatile profiles, and the evolution of an increased number of volatile compounds during processing was monitored by GC-MS-based fingerprinting method. Finally, chemometrics was adopted to help understand the key patterns and identify discriminant compounds. Overall, this is a complete study with solid data and good logic. However, some concerns are needed to be addressed. Therefore, I recommend a major revision prior to publication.

  1. What is the purpose of this work? It is not that clear. Please clarify it in the abstract.

We have added a sentence clarifying the fact in the abstract (Lines 13-14) as "The purpose of this work was to understand processing time's effect on legume's volatile profile."

  1. It is mentioned that legumes are not as fully utilized (partly) due to volatile compounds contributing to their undesirable odour. So, how do you define the undesirable odour? Is it a smell undesirable to a vast of people or to a small group ones?

We have clarified that it is to a vast proportion of people in the introduction (Line 36).

  1. Actually, the factorsinvolved in cooking time of beans are diverse and complex, why do the authors choose 95 °C in the experiments?

The authors agree that the factors affecting beans are indeed diverse. The authors chose 95 °C, representing a temperature most closely related to common boiling (Line 85). Though literature had chosen this temperature before, a step-by-step increase of duration from 0 minutes to 120 minutes had never been done before. Furthermore, choosing this temperature allowed us to tightly control each processing treatment.

  1. Do other factors like environment, post-harvest storage and preprocessing influence the cooking time?

Yes, the mentioned factors would influence the cooking time. The authors minimised those effects by ensuring that all legumes used in the experiment were of the same batch purchased from the same commercial source. Without sunlight, post-harvest storage was controlled at 4° C (Line 95). Pre-processing was controlled at 16 hours of soaking time and 20° C in an incubator without light (Line 109).

  1. It seems that this work aims to give consumers and food industry valuable information on selecting processing intensity to maximise legume utilization. Why only the hydrothermally processed method was discussed rather than other food processing methods like smoking or picking?

Thank you for the valuable suggestion. The authors chose to focus on hydrothermal processing (opened versus closed) because it is the method most accessible by consumers (opened/home boiling), and it would approximate the state consumer might purchase them in (closed/canning) – Line 84 - 86. Certainly, the authors acknowledged that studying other less common methods like pickling or smoking will also yield valuable insight to improve legume utilisation.

We have addressed this important aspect as follows in the revised manuscript: "Moreover, understanding the effect of other non-hydrothermal processed methods such as smoking and pickling on the legume's volatile profile could be considered in forthcoming studies." (Lines 572-574).

  1. How do you evaluate the odour? By sensory system or just by human nose? How do the authors judge the desirable level of odour?

Certainly, odour evaluation of the legumes will further inform consumer and industrial processing methods. Unfortunately, the current study only evaluated the (instrumental) volatile profile, not the odour threshold that humans would perceive. The collected information is only indicative of the change in the volatile flavour profile, and the odour relevance of the selected discriminant compounds was discussed based on literature information. We also can not discuss the desirable odour level based on our instrumental volatile fingerprinting data. This is reflected by lines 571 – 572: "In the future, there is a need for sensory analysis to characterize and quantify the odour attributes and also establish a desirable odour level."

Reviewer 2 Report

In my opinion, this research paper contains significant and relevant information that justify publication, with minor revision regarding the following issues:

1.     References:

a)       References citation should be carefully revised to use the style guidelines of Molecules: “ In the text, reference numbers should be placed in square brackets [ ], and placed before the punctuation; for example [1], [1–3] or [1,3].” For example,  in page 1 line 31 the citation (Tharanathan & Mahadevamma, 2003), should be cited as [1].

b)       References section should be revised in order to follow the Instructions for Authors: “ References must be numbered in order of appearance in the text (including table captions and figure legends) and listed individually at the end of the manuscript.. Please revise this aspect. For example, the reference (Tharanathan & Mahadevamma, 2003), corresponding to the reference presented in page 24 line 702, should be in page 21 line 606 as follows:

 1. Tharanathan, R. N., & Mahadevamma, S. (2003). Grain legumes—a boon to human nutrition. Trends in Food Science & Technology, 702 14(12), 507-518. doi:https://doi.org/10.1016/j.tifs.2003.07.002

2.     Abstract – page 1 line 12- removed he redundancy “(partly)”

3.     Introduction

The content is succinctly described and contextualized with respect to previous and present theoretical background on the topic and supported by relevant references on the topic, although relevant recent references on the topic are missing, such as:

-        Christian Trindler, Katrin Annika Kopf-Bolanz, Christoph Denkel, 2022. Aroma of peas, its constituents and reduction strategies – Effects from breeding to processing, Food Chemistry, Volume 376, 131892, doi.org/10.1016/j.foodchem.2021.131892.

-        Aravindakshan, S.; Nguyen, T.H.A.; Kyomugasho, C.; Buvé, C.; Dewettinck, K.; Van Loey, A.; Hendrickx, M.E. The Impact of Drying and Rehydration on the Structural Properties and Quality Attributes of Pre-Cooked Dried Beans. Foods 2021, 10, 1665. https://doi.org/10.3390/foods10071665

-        Bassett, A., Kamfwa, K., Ambachew, D. et al. Genetic variability and genome-wide association analysis of flavor and texture in cooked beans (Phaseolus vulgaris L.). Theor Appl Genet 134, 959–978 (2021). https://doi.org/10.1007/s00122-020-03745-3

 4.     Results

1)       Suggestion: present results of “ 3.1.Starch, protein, lipid and moisture content of cowpea, chickpea and kidney bean”, (page 5 – line 227-230), in table format.

2) In section 3.3 (page 6 lines 277 – 279),  and (page 7 , lines  280-281, lines  293-295, lines 299-301,  lines 305-306, information is not clearly supported, is it possible to present data in a different format for better understanding?  Some information in supplementary materials, may be presented in this section?

Author Response

Dear, 

The authors thank the reviewer for providing valuable feedback in this peer review process. Please find each reviewer's point of comment and the author's explanation below.

Best regards, Biniam Kebede

In my opinion, this research paper contains significant and relevant information that justify publication, with minor revision regarding the following issues:

  1. References: a)      References citation should be carefully revised to use the style guidelines of Molecules: "In the text, reference numbers should be placed in square brackets [ ], and placed before the punctuation; for example [1], [1–3] or [1,3]." For example,  in page 1 line 31 the citation (Tharanathan & Mahadevamma, 2003), should be cited as b)      References section should be revised in order to follow the Instructions for Authors: "References must be numbered in order of appearance in the text (including table captions and figure legends) and listed individually at the end of the manuscript.". Please revise this aspect. For example, the reference (Tharanathan & Mahadevamma, 2003), corresponding to the reference presented in page 24 line 702, should be in page 21 line 606 as follows:
  2. Tharanathan, R. N., & Mahadevamma, S. (2003). Grain legumes—a boon to human nutrition. Trends in Food Science & Technology, 702 14(12), 507-518. doi:https://doi.org/10.1016/j.tifs.2003.07.002

The authors thank the reviewer for clarifying this point. References in the revised manuscript have been updated to meet the Guide for Authors of MDPI.

  1. Abstract – page 1 line 12- removed he redundancy "(partly)" 

The word "partly" has been removed as per the suggestion (see Line 12).

  1. Introduction

The content is succinctly described and contextualized with respect to previous and present theoretical background on the topic and supported by relevant references on the topic, although relevant recent references on the topic are missing, such as:

-        Christian Trindler, Katrin Annika Kopf-Bolanz, Christoph Denkel, 2022. Aroma of peas, its constituents and reduction strategies – Effects from breeding to processing, Food Chemistry, Volume 376, 131892, doi.org/10.1016/j.foodchem.2021.131892.

-        Aravindakshan, S.; Nguyen, T.H.A.; Kyomugasho, C.; Buvé, C.; Dewettinck, K.; Van Loey, A.; Hendrickx, M.E. The Impact of Drying and Rehydration on the Structural Properties and Quality Attributes of Pre-Cooked Dried Beans. Foods 2021, 10, 1665. https://doi.org/10.3390/foods10071665

-        Bassett, A., Kamfwa, K., Ambachew, D. et al. Genetic variability and genome-wide association analysis of flavor and texture in cooked beans (Phaseolus vulgaris L.). Theor Appl Genet 134, 959–978 (2021). https://doi.org/10.1007/s00122-020-03745-3

The authors gratefully acknowledged the recent literature presented by the reviewer. Paper one [Reference #11] has been added as a reference describing lipid oxidation as a source of (undesirable) volatile compounds (Lines 53 and 273). Paper two [Reference #8] and three [Reference #9] have been added as supporting references in Line 46.

  1.  Results
  • Suggestion: present results of "3.1.Starch, protein, lipid and moisture content of cowpea, chickpea and kidney bean", (page 5 – line 227-230), in table format.

The authors thank the reviewer for the suggestion. We have chosen to retain the original format to be in line with the literature investigating legumes. Furthermore, since the result table would not be presented as a complete proximate composition table, we fear it may confuse the readers when the percentages do not add up to 100%. In the future, if the proximate composition is done, we will indeed put them in a table format.

2) In section 3.3 (page 6 lines 277 – 279),  and (page 7 , lines  280-281, lines  293-295, lines 299-301,  lines 305-306, information is not clearly supported, is it possible to present data in a different format for better understanding?  Some information in supplementary materials, may be presented in this section?

As suggested, the authors have created a figure to better illustrate the improvements in volatile compound detection between the opened and closed systems (Figure 1, Line 287). Furthermore, Figure 1 also visually illustrates the chemical class information presented in lines 277 – 306.

Reviewer 3 Report

I suggest some revisions and recommendations to your manuscript before re-considering publication in the attached file. 

Author Response

Dear, 

The authors thank the reviewer for providing valuable feedback in this peer review process. Please find each reviewer's point of comment and the author's explanation below.

Best regards, Biniam Kebede

 The following comments to be addressed from the authors:

-Line 19: Please specify the effect of the treatment over 30 min on volatiles and fatty acids? significant, or no?

The sentence has been modified to show a significant change in volatile profile but no significant change in fatty acid.

Lines 19-22: "This work showed that processing at 95°C for 30 minutes significantly reduced the amount of compounds commonly associated with undesirable odour, but showed no significant change in fatty acid profile."

-Line 14: volatile and fatty acids profile! write profiles.

We have revised as 'profiles' (Line 15).

-Line 14: The soaked samples? are you state the soaking process in the objectives (in introduction)?

We did not study the sole effect of soaking on the sample. Before hydrothermal processing, all legumes were pre-soaked (see the full description of materials and methods in Section 2.3). We have clarified this in the abstracts as "All legumes were pre-soaked (16 h) and then hydrothermally processed at 95°C for 15 to 120 min…." (Lines 15-18).

-Line 36: The lower protein digestibility is due to natural barriers in the plant cell and antinutrients such as enzyme inhibitors (Carbonaro, Grant, Cappelloni, & Pusztai, 2000)! could you write other examples of antinutrients in legumes e.g., saponins, tannins, phytic acid, etc., as well as cite more recent reference.

The authors thank Reviewer #3 for the suggestion. We have revised as follow: "… antinutrients such as enzyme inhibitors, tannins and phytic acid [3,4]." (Lines 38-39) to show the non-exhaustive nature of antinutrients presented and included a more recent supporting reference.

-Line 60- 86: I noticed there are several old references, please update with new ones if possible.

Yes, the references are from a number of years back (such as the cowpeas' reference), as it reflects the lack of literature in recent years on the volatile compounds of legumes (Line 61-62).

-line 51: check the spelling of (un)cooked legumes.

Thank you for the suggestion. The authors clarify that the spelling is intentional to indicate both uncooked and cooked states in a succinct sentence (Line 51).

-line 91: different processing time?? please specify exactly the processing i.e., time and temperature conditions, including both open and closed hydrothermal.

The authors have added: "…processing time (up to 120 min)." in Lines 86-97. The exact condition is thereafter detailed in the Materials and Methods Section 2.3.

-Line 91: (The impact 91 of the process will be investigated on the fatty acid and volatile profiles) could you please merge this phrase into line 88, so the aim should be in one sentence in order to matched well with the title.

Thank you. The sentence has been merged as suggested; see Lines 87 onwards.

-line 98: the title 2.1. Sample preparation and storage is insufficient because you include under this title other concept (proximate analyses i.e., starch, protein, lipids, and moisture analyses)? Thus, you can make a separate subtitle related to proximate analyses. But I think you can write Compositional analyses since you didn't analyze all proximate composition like carbohydrates, ash and fiber. Also, you should write compositional analyses of "uncooked cowpea, chickpea and kidney bean"

The changes have been implemented as suggested. Please see Lines 100 onwards, and 2 separate headings have been created: "2.2. Compositional analyses of uncooked cowpeas, chickpeas and kidney beans" and "2.3. Hydrothermal processing of cowpeas, chickpeas and kidney beans".

-Line 100: were purchased from a local market in Dunedin in August 2017? I proposed that the sampling date is very old? What are the storage conditions before 2017 labeled on the samples? What is the production date of these legumes before 2017? Please check? To be ensure the samples not expired according to Global Food Associations or Standards. e.g., According to the United States Agency for International Development, USAID "After 2–3 years, the beans will start losing their nutritional value, and most naturally found vitamins will be gone within 5 years".

We thank the reviewer for raising your concern. The authors are aware of this degradation during storage and have stored them with many precautions, such as using opaque aluminium bags, vacuum sealed and stored at 4 °C (Line 97). The authors would like to clarify that the actual experiments (including cooking and all chemical analysis) were conducted between late 2018 and early 2019, though dissemination of the results has been delayed. This information has been added to Line 98. The storage condition before the purchase was in a sealed plastic bag, as commonly available legumes were sold. Physical inspections and discarding of deformed and damaged seeds prevented experimentation with possible seeds with reduced nutritional value. Unfortunately, the physical package and label could not be recovered for checking.

Finally, the research aims to take "non-thermal processed" samples within the same batch so that the effect of processing time can be investigated as a single factor. The authors also incorporated chemometrics into the statistical analysis (Section 2.7 Data analysis, Line 206). Thus, while the authors acknowledged that the point raised by the reviewer is a potential weakness of the experiment, through physical inspections of the seeds, handling of the materials, investigation procedures and statistics, we are confident that the results of the experiment were not compromised.

-Line 107: write the date of reference AOAC Official Methods.

Year of method usage added (2017). Please see Line 104.

-Line 109- 110: 2.2. Hydrothermal processing of legumes! Please write the 3 legumes names specifically no write legumes in general. Apply throughout the manuscript. (Write: Hydrothermal processing of cowpea, chickpea and kidney bean).

Instances of sentences that the reviewer mentioned have been revised throughout the manuscript.

- Line 130: please convert 3,000 rpm to x g unit.

The authors thank the reviewer for the suggestion. The authors have chosen to retain the original wording, as the mechanism of action of an Ultra Turrax homogeniser was to homogenise the sample rather than using this equipment for centrifugal action.

-Lines 120 & 134: Why you stored the treatments at various conditions (-81 °C and -20 °C)?

The storage of samples at -20 °C and -81 °C were due to the limitation of the analytical instruments. The sample stored at -81 °C had a longer lag time before analysis (~3 weeks), whereas the sample stored at -20 °C had a shorter period of lag time (~2 days) before analysis. Ideally, the samples would be preferably stored at -81 °C, but a limitation on the storage space necessitated this action.

-line 127, and in line 15: for varying durations (15, 30, 45, 60, 90 and 120 min), Is zero-time has been carried out as control? (Please include 0 min in abstract and methodology as well) I see 0 time in Table 1, therefore please revise this concern throughout the manuscript. In line 120 you stated that 15 may be act as control to see the trends of volatile compounds change comparing with gradually increasing time, please revise.

Revision has been done according to the suggestion to clarify that 0 min (not heated) was included in the experiment (Lines 125-126). To clarify, we did not use 15 min of cooking as control.

We have added the following sentence in the revised manuscript: "The tube containing pre-soaked legumes was not hydrothermally processed to represent a time of 0 min (i.e. control of the cooking experiment)." (Lines 125-126).

-Line 136: Determination of fatty acids in legume seeds using FAME-GC-FID? Check the method? You could write: using chromatography flame ionisation detection (GC-FID). Also, don't write abbreviation in titles or subtitles.

The heading title has been changed according to the reviewer's suggestion as "2.4. Determination of fatty acids in cowpeas, chickpeas and kidney beans using chromatography flame ionisation detection" – please see Line 134.

-Line 140: write the thawing temperature? As done in line 164.

Line 139 has been modified to include thawing at 4 °C for 12 hours.

Line 160: revise the title? You can write: Determination of volatile compounds using Headspace solid-phase micro-extraction gas chromatography-mass spectrometry.

The heading title has been revised according to the reviewer's suggestion as "2.5. Determination of volatile compounds using headspace solid-phase micro-extraction gas chromatography-mass spectrometry" – please see Line 159.

-Line 242: add "cooked" before cowpeas….

Added cooked in brackets (cooked) before cowpeas to acknowledge uncooked samples as well (see Lines 241).

-Line 323: replace as a function to as function.

The "a" has been removed (Line 326).

-line 325: replace the volatile fraction to the volatile fractions.

The "s" has been added as suggested by the reviewer (Line 328).

- In line 111 you wrote (legumes (500 g) were soaked), thus all processed samples at various time previously were soaked? On the other hand, in lines 338 and 342, you wrote soaked sample only has been done for 0 time? Please revise that concern.

Yes, all processed samples treated at various times have been pre-soaked beforehand. Line 340 referred to the wording SOAK on the figure, not that only that particular sample was soaked. This has been clarified in the manuscript (Line 340).

-Specific comments:

- For sub titles 2.3. and 2.4. you should write "cooked" before the legume's samples, apply throughout the manuscript.

-Revise all subtitles to reflect the text content precisely

Changes as has been made according to the reviewer's comments. The author respectfully informs the reviewer that the "cooked" wording was not added in front of samples, as the materials and methods include samples which are only soaked but not cooked as well.

- The keywords are long, thus focus only up 6 from most relevant keywords.

We respectfully choose to keep the current keywords, as the authors are of the opinion that reducing the keywords could not fully capturing the scope of the work well.

-Please follow the journal references format.

The reference format has been updated according to the journal's guidelines.

-Make p < 0.05 italic "p < 0.05" and uniform their style, sometimes with space, while other place with space.

We have made the suggested changes throughout the manuscript.

Round 2

Reviewer 1 Report

Since my concerns have been addressed, I suggest the work is acceptable in present form.